# Hesitations and Aspirations of Farmers in Nature-Protected Areas

Angela Turck * and Wiltrud Terlau

International Centre for Sustainable Development (IZNE), Bonn-Rhein-Sieg University of Applied Sciences, 53757 Sankt Augustin, Germany
* Correspondence: angela.turck@h-brs.de

**Abstract:** Pursuant to Sustainable Development Goal (SDG) 15 of the 2030 Agenda for Sustainable Development of the United Nations, one pivotal target is to halt biodiversity loss. This paper's objective is to analyze why and how German farmers hesitate to implement more than the prescriptive measures with regard to cross compliance and direct payments under the European Common Agricultural Policy (CAP) and what their aspirations are for possible incentives to bring biodiversity into focus. By applying a mixed methods approach, we investigate the experience of individual farmers by means of a qualitative approach followed by a quantitative study. This analysis sheds light on how farmers perceive indirect influencing factors and how these factors play a non-negligible role in farmers´ commitment to biodiversity. Economy, policy and society are intertwined and need to be considered from a multi-faceted perspective. In addition, an in-depth analysis is conducted based on online focus group discussions to determine whether farmers accept financial support, focusing on both action- and success-oriented payments. Our results highlight the importance of paying attention to the heterogeneity of farmers, their locations and, consequently, farmers' different views on indirect drivers influencing agricultural processes, showing the complexity of the problem. Although farmers' expectations can be met with financial allocations, other aspects must also be taken into account.

**Keywords:** biodiversity; farmers' heterogeneity; incentives; Sustainable Development Goals (SDGs); mixed methods approach

## 1. Introduction

One of the 17 Sustainable Development Goals (SDGs) set by the United Nations (UN) in 2016 is SDG 15 (Life on Land) [1]. Target 15.5 states that there is a need for "urgent and significant action to reduce the degradation of natural habitats, halt the loss of biodiversity and, by 2020, protect and prevent the extinction of threatened species" [2]. Insects, the most species-rich class of animals, and thus an essential component of biological diversity, can be found in almost every habitat on land and water. Insects contribute to the nutrition of other animals and humans as part of our world´s food production, notably through pollination. Furthermore, many insects themselves serve as food for other animals and are helpful for pest control [3]. In 2017, a publication on the "Insect biomass decline" by Hallmann et al. displayed data on a worrying drop in insects [4]. This also applies to insects in nature-protected areas [5–7]. An indicator of extinction of threatened species is the IUCN (International Union for Conservation of Nature) Red List of Threatened Species, which is also known as the IUCN Red List [8]. This list displays an inventory list of the global conservation status of different species, including insects. The latest Red List of the German Federal Agency for Nature Conservation was published in 2021/2022, which confirmed advanced decline in insects. Almost 30% of insects are classified as endangered, critically endangered or endangered to an unknown extent [9].

Nature-protected areas are legally binding areas in Germany where the special protection of nature and landscape is required. A regulation for such areas can be found in the

German Federal Nature Conservation Act in § 23 (1) [10], and within these nature-protected areas, owners must accept restrictions that come with the declaration of protection, pursuant to § 14 (2) of the German Basic Law [11]. Since this is a perceived encroachment on the rights of third parties, disputes may arise between owners and nature conservation authorities, especially with regard to said restrictions. As farming is practiced in nature-protected areas, the Common Agricultural Policy (CAP) of the European Union (EU) also affects farmers working on arable land within nature-protected areas [12]. The CAP has been one of the most important political areas of European policy since the establishment of the European Community in 1957 and has been repeatedly adjusted to the living situations in Europe over the past decades [13,14]. Globalization, climate change and the strengthening of rural areas are central regulatory aspects of the CAP [14]. In this context, agriculture is caught between social and ecological responsibility as well as the economic necessity for sustainable entrepreneurial action.

Farmers receive direct and indirect payments. These subsidies are bound to conditions such as food safety, animal welfare and environmental protection. In addition to EU funds, German farmers receive federal and state subsidies, such as payments from agri-environmental programs. The EU agricultural support is divided into the first and second pillars based on binding and voluntary measures, respectively. The direct payments outlined by the CAP are found in the first pillar. They are the core element of EU agricultural support. This instrument supports the income and protects agricultural enterprises from risk, regardless of production. These payments are granted per hectare of agricultural land— if the respective conditions are met. The linking of agricultural payments to obligations concerning environmental, human, animal and plant health as well as animal welfare is known as "cross compliance". Another payment outlined in this first pillar goes towards climate and environmentally friendly land management practices, which is defined as "greening". "Greening" promotes agricultural services for climate protection, species conservation, diverse cultural landscapes and sustainable production [15]. The second pillar comprises targeted support programs for sustainable and environmentally sound management, as well as for rural development. Especially for farmers, the focus is on voluntary agri-environmental and climate protection measures (AECM) in agriculture, in addition to payments for organic farming and for "Natura 2000" sites [16].

Having implemented these regulations, farmers in Germany feel trapped in the so-called "trilemma" [17]. Farmers are not only obligated to counteract climate change and provide food security, but legislation also requires farmers to enhance the biodiversity of their farmed land. The farmers' ability to think economically is disturbed by political framework conditions set by the EU ("Green Deal" [18] and "Farm to Fork Strategy" [19]) and Germany. Multiple regulatory requirements exist [20]; some of them are incomprehensible to farmers, which leads to growing discontent among these farmers. In addition, the diverse societal expectations towards farmers' management of land use and society's eagerness not to pay the additional cost of such "services to environment" come into conflict. An important keyword here is "consumer behavior" [21], which resonates poorly with farmers. The farmers have clearly expressed their discontent through their demonstrations in 2019. In this context, the German Diversity of Insects in Nature-protected Areas (DINA) project, a trans- and interdisciplinary research study, was started [22]. The DINA project is based on an interactive approach to integrate scientific findings with socio-ecological aspects and stakeholder perspectives. The socio-ecological aspects of the project includes a stakeholder analysis to identify farmers working on arable land within nature-protected areas as key stakeholders [23]. Nowadays, farmers face a variety of challenges such as drought, energy shortages, and restrictions on fertilization due to the Plant Protection Act and Plant Protection Application Ordinance. They are components of the Insect Diversity Protection Act introduced in September 2021 [24]. Farmers have to manage all this and more. In order to gain insight into the hesitations and aspirations of farmers, we conducted a qualitative study [23] followed by a quantitative survey using previous findings that again will later be complemented by a further qualitative study. This research centers on

the question of how farmers cope with the conservation of biodiversity on arable land within nature-protected areas. Obstacles and possible financial incentives, inter alia, are examined. The focus on arable land in such protected areas is chosen because the protection of this land is not anchored in the current German Arable Farming Strategy 2035 (Ackerbaustratgie 2035) [25]. With our chosen approach we gain an understanding of farmers' commitment to promoting biodiversity by examining constraints on and aspirations of farmers in nature-protected areas. We aim to determine whether financial incentives are sufficient to motivate farmers to enhance biodiversity in nature-protected areas. As in the past, the world political situation was different from the one we find today.

## 2. Methodology and Research Findings

The combination of qualitative and quantitative research methods, the so-called "mixed methods design" approach, is a method that has received more attention in the past several years [26–28]. The used data collection tools were applied in three consecutive steps [29]. Firstly, our data were generated with a qualitative study [30] which supported the subsequent design of the quantitative (second) study that, in turn, led to a later qualitative (third) study [31]. This kind of triangulation promises a deeper insight into farmers' obstacles in promoting biodiversity. Because of the dynamic momentum of the changing political and economic situations, generating knowledge is essential.

The collected data were consistently anonymized.

### 2.1. Qualitative Study (First Stage)

With the help of a qualitative study that targeted and identified key stakeholders, i.e., participating farmers who work in 21 nature-protected areas within our DINA project, 33 farmers were asked to participate in a semi-structured questionnaire [30]. One goal was to learn about their concerns and to understand their scope of action, i.e., their decision-making framework. Our evaluation shows that agricultural production is influenced by factors and direct and indirect drivers, which confront farmers with complex decision-making situations [32]. Direct drivers include land use and agricultural production as well as habitat condition and quality. In contrast, indirect drivers (i.e., economy, policy and social frameworks) are beyond the immediate control of individual farmers [7]. Farmers are required to make decisions on a daily basis, and they have to find a balance between these factors [23]. Economically, they are confronted with the erosion of price and increased production demands, leaving little space for considering ecological aspects such as the environment and biodiversity. This is what we call the "dilemma of farmers" [23]. Moreover, the land use stems from multiple demands made on the land: for the benefit of climate change, securing food and enhancing biodiversity. The German Advisory Council on Global Change (WBGU) defines this as the "trilemma" of land use [17]. In the middle of the trilemma are the farmers, as demonstrated in Figure 1.

Therefore, it can be determined that the indirect influencing factors, the framework conditions, are a nuisance to farmers. This led us to focus on indirect drivers in the later quantitative study.

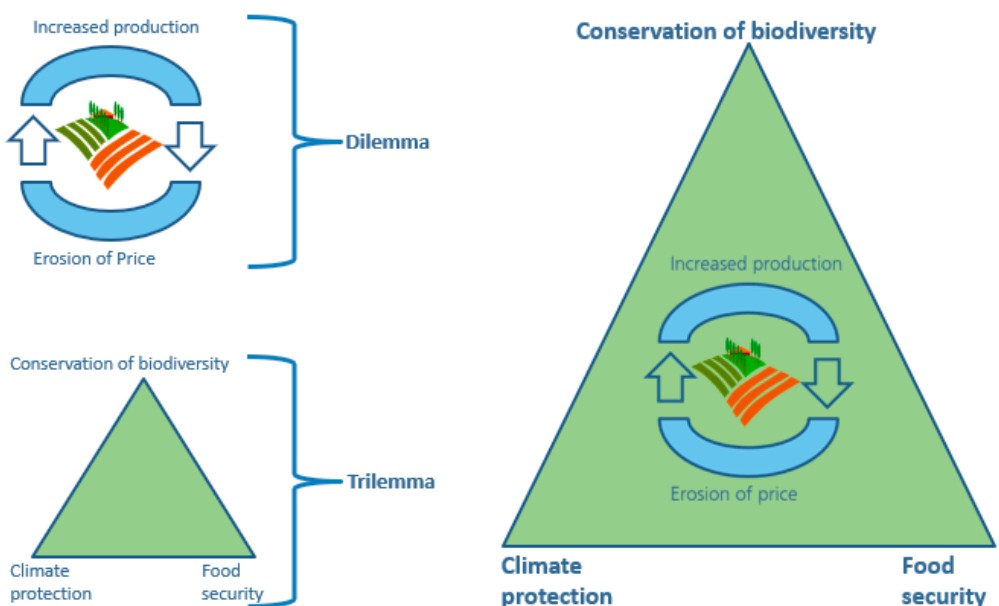

**Figure 1.** Farmers´ dilemma and trilemma—a schematic illustration (source: author´s elaboration based on [17].

### 2.2. Quantitative Study (Second Stage)

The gained insights obtained by the qualitative study were used to further refine our findings. The quantitative survey (second stage) which followed consisted of a computer-assisted telephone interview (CATI) and an in-depth and subsequent qualitative study, the latter being an online focus group study (third stage). This methodology of using a three-stage approach promises a deeper understanding of farmer´s concerns. The German polling institute dimap was commissioned to carry out both steps [33].

#### 2.2.1. Computer-Assisted Telephone Interviews (CATI) and the Research Findings

CATI Study: The Basic Population of the Study Comprised Farmers Who Cultivate Arable Land or Grow Wine or Fruit in Nature-Protected Areas. The sample was selected via pre-selection of the areas by the project partners. From the 100 nature-protected areas in Question that were of interest to the DINA Project, 97 farmers were interviewed. The duration of an interview was about 20 min, and the survey period was from 27 December 2021 to 2 February 2022.

An interview in its various forms remains the main method of practical social science [34]. However, an interview can only guarantee a sufficiently thorough amount of knowledge if it is conducted in a controllable form [35]. The technology of computer-assisted telephone interviews enables the efficient handling of telephone surveys [36]. The ability to target calls by geographic regions, the possibility that the respondent can ask to clarify questions and the substantially low attrition rate ensure these aspects [33]. The selected telephone survey method is a reliable instrument of social research, and the interviewer-assisted survey provides quality assurances during the survey process. The telephone interview was conducted using the questionnaire displayed on the screen of the computer. The questionnaire was developed to achieve coherent and comparable interviews [37]. The interviewer recorded the given answers of the interviewee by using the keyboard or mouse, which matched with the pre-coded answers displayed on the screen. In our study, farmers were notified and informed about the objectives of the survey beforehand by means of a letter and informative brochure. The data of farmers operating in 100 selected German nature-protected areas were included. The areas were selected in consultation with project partners, and according to the same method, were used for selecting the 21 DINA project areas [22,38]. The response rate was 97% (*n* = 97).

Based on previously obtained knowledge, 35 questions in a sensible order as well as 5 mandatory statistical questions for possible later evaluation were developed.

The following observations are selected results from the telephone survey. Figure 2 shows that 96% of the surveyed farmers indicated restrictions on their land use within nature-protected areas. The respondents named these restrictions in an open question as requirements for plant protection, fertilizers, growth regulators, herbicides and a period of non-cultivation (fallow). Rarely mentioned conditions, on the other hand, included leaving stubble cereals standing, double seed row spacing, the use of regional seed and crop rotation requirements or harvest waiting periods, such as partially leaving crops or alternating crops left standing. For the query about receiving compensation, 59% of the respondents do not receive any payment, whereas almost 41 % receive compensation.

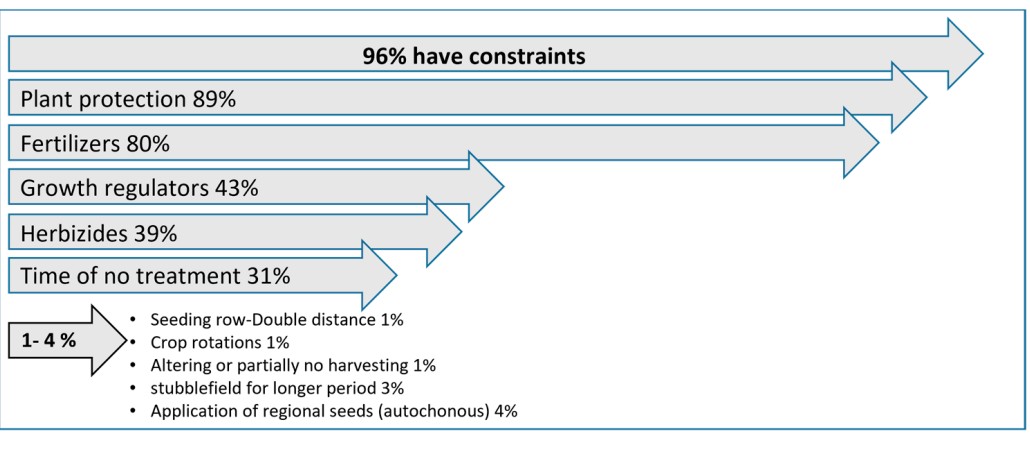

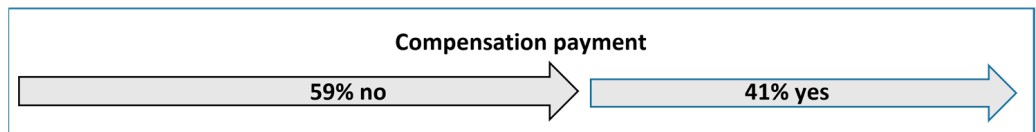

**Figure 2.** CATI-selected results: restrictions and received compensation payment (*n* = 97).

Table 1 gives an overview of payments received, which vary considerably. The mean value is EUR 450 per hectare and per financial year, and the median is EUR 300; additionally, the minimum value is EUR 40 and the maximum is EUR 1.600.

**Table 1.** CATI-selected results: amount of payment. Data in absolute and relative frequencies (*n* = 39).

| Received Payments (EUR/ha) | Absolute Frequencies (Relative Frequencies) | Received Payments (EUR/ha) | Absolute Frequencies (Relative Frequencies) | Received Payments (EUR/ha) | Absolute Frequencies (Relative Frequencies) |
|---|---|---|---|---|---|
| <100 | 2 (5.1%) | 250–299 | 5 (12.8%) | 500–599 | 2 (5.1%) |
| 100–149 | 2 (5.1%) | 300–349 | 4 (10.3%) | 600–699 | 4 (10.3%) |
| 150–199 | 4 (10.3%) | 350–399 | 1 (2.6%) | ≥700 | 7 (18.0%) |
| 200–249 | 4 (10.3%) | 400–499 | 3 (7.6%) | Do not know | 1 (2.6%) |

In the case of farmers working in nature-protected areas, monetary needs are a vital factor. Farmers were invited to state their assessment of lost profits or perceived losses due to restrictions on land use. The question aimed to understand the subjectively felt reduction in acquisition opportunities due to existing usage requirements. Overall, more than 80% of the participants complained about a loss of profit. The perceived average loss is EUR 895 per hectare, whereas the median is EUR 400 per hectare, with a range from EUR 100 to EUR 8000 per hectare (Table 2).

**Table 2.** CATI-selected results: perceived losses and estimation of monetary loss per hectare per financial year of respondents. Data in absolute and relative frequencies (*n* = 49).

| Estimated Monetary Loss (EUR/ha) | Absolute Frequencies (Relative Frequencies) | Estimated Monetary Loss (EUR/ha) | Absolute Frequencies (Relative Frequencies) | Estimated Monetary Loss (EUR/ha) | Absolute Frequencies (Relative Frequencies) |
|---|---|---|---|---|---|
| <200 | 6 (12.2%) | 400 < 499 | 7 (14.3%) | ≥3000 | 4 (8.2%) |
| 200–299 | 7 (14.3%) | 500 < 999 | 10 (20.4%) | Do not know | 3 (6.1%) |
| 300 < 399 | 7 (14.3%) | 1000 < 2999 | 5 (10.2%) | | |

This scenario is followed by a question on whether farmers can imagine biodiversity having absolute priority in their land management, provided they receive monetary compensation. This was answered positively by 65% of respondents (Figure 3).

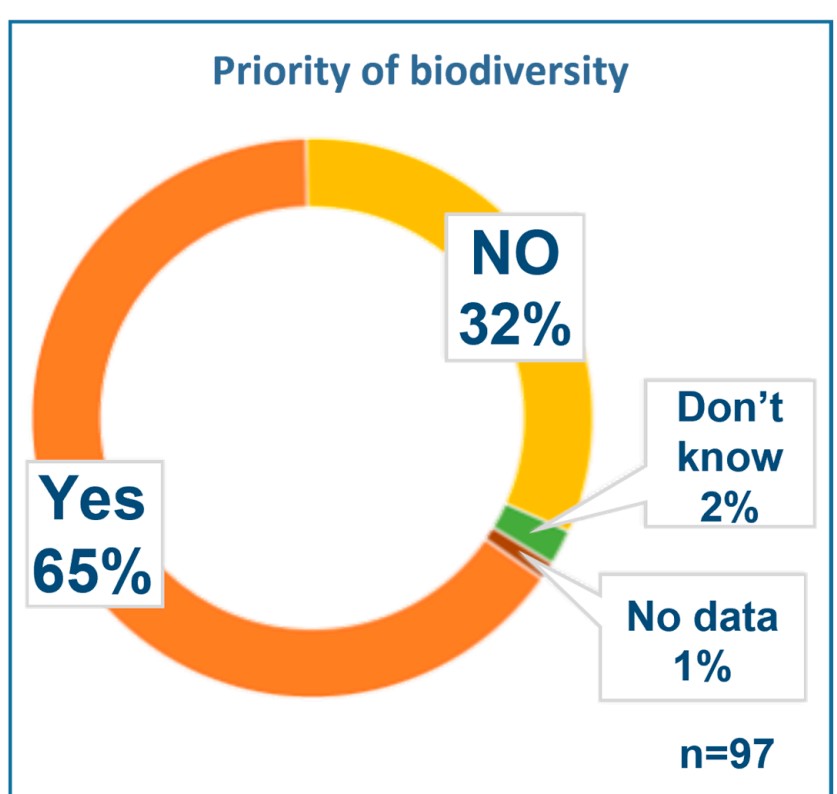

**Figure 3.** CATI-selected results: biodiversity is a priority provided monetary compensation is received. Data in relative frequencies (*n* = 97).

Table 3 shows farmers´ monetary wishes for prioritizing biodiversity. The desired necessary financial incentive component, i.e., amount, ranges from EUR 300 to EUR 7500 per ha per financial year. It is remarkable that more than 25% of the respondents could not or did not want to give any information related to this topic.

**Table 3.** CATI-selected results: desired monetary incentives per hectare for prioritizing biodiversity. Data in absolute and relative frequencies (*n* = 97).

| ≤EUR 500 | 9 (9.3%) | EUR 2000–EUR 2999 | 5 (5.2%) | Do Not Know | 16 (16.5%) |
|---|---|---|---|---|---|
| EUR 500–EUR 999 | 26 (26.8%) | ≥EUR 3000 | 5 (5.2%) | No data | 12 (12.4%) |
| EUR 1000–EUR 1999 | 24 (24.7%) | | | | |

Monetary incentives are a crucial factor in motivating farmers to increase biodiversity. Moreover, the literature distinguishes and discusses how payments that reward not only action, but also success (outcome-based agri-environmental measures), may be an additional suitable approach [39–42]. As Figure 4 shows this question was answered in the affirmative by around 75% of the farmers surveyed.

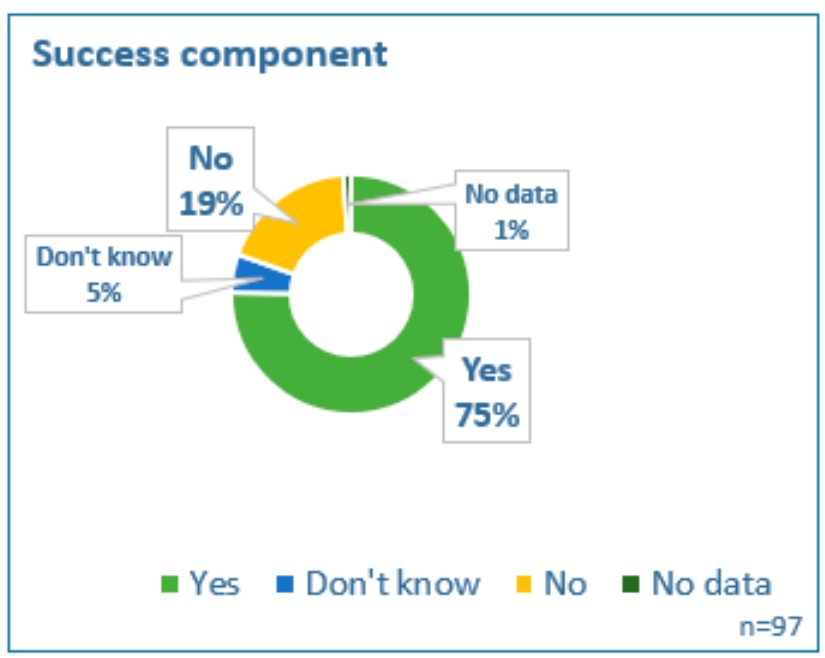

**Figure 4.** CATI-selected results: acceptance of outcome-based payments. Data in relative frequencies (*n* = 97).

For the sake of completeness, it is to be mentioned here that other questions in the telephone survey included facts about hourly wages, time spent working in nature-protected areas and the farmers' assessment of the effort they put into biodiversity measures at their desks.

In order to take a closer look at the option of a success fee, online focus group interviews, the third stage of the study, were carried out.

### 2.2.2. Qualitative Study of Online Focus Groups (Third Stage) and the Research Findings

Generated from the aforementioned quantitative study, we continued the survey of farmers in greater depth with the help of a qualitative focus group study. The methodological approach of focus groups (here, online focus groups) is a diagnostic tool that reveals the emotional and rational anchored attitudes of the target group (i.e., farmers working on arable land in nature-protected areas). The results are not meant to be representative in a statistical sense. Instead, focus group studies collect a wide spectrum of different experiences, perspectives, feelings, ways of thinking and evaluations with regard to a transparent topic to identify typical attitudes. An interview guideline was developed.

A total of 15 of the former 97 participants of the previous CATI study, who are farmers who manage land in nature-protected areas in Germany, dicussed, among other things, how they currently mangage their land in the protected areas. The online focus group interviews took place on 6, 7 and 12 April 2022. Each discussion round had three to five participants and a duration of about 120 minutes. The participants were divided into four online focus groups and allocated a time based on their time preference; no other criteria were applied.

As in the preceding telephone interviews, we commissioned the opinion research institute dimap to lead the focus groups [33]. The aim of the online dialogues was to capture the interpretation, views and attitudes of the respondent farmers in discussion.

They were recorded and subsequently transcribed. Using a content analysis, an often practiced form of qualitative data analysis working with categories and the categorization of the material [43,44], we focused on monetary incentives for encouraging biodiversity in nature-protected areas, as well as on the potential acceptance of success-oriented payments. Statements were coded into categories. As the coding process was carried out and the participants were promised anonymity, a summary of statements are presented rather than word-by-word quotations.

The current financial compensation for participation in biodiversity-enhancing measures based on action was mostly described by farmers as inadequate and was criticized for not reflecting the high value placed on biodiversity. Additionally, the payment of lump sums (flat rates) was predominantly viewed by the respondents in a negative light, and they believe that location and yield quality, as well as additional expenses and cost increases, are not accounted for. Furthermore, farmers advocated minimizing administrative expenses, as they state that their perceived burden is immense. With regard to conceivable models for creating monetary incentives for biodiversity-friendly management, the farmers unanimously advocated for a reward system. However, unlike the responses from the farmers during the telephone interviews, the farmers participating in the online focus groups had a critical view of the additional payment of result-based (success) fees or bonuses. They affirmed this by calling into doubt the practical design of targets and performance controls. They believe that it would be challenging to implement in view of the work in and with nature. The farmers state that effectiveness would be difficult to measure, as other environmental influences are relevant and, in addition, a singular consideration of small areas is not possible and not comparable.

Irrespective of financial aspects, most farmers agreed that measures to promote biodiversity can and will be implemented by them, provided that they are not linked to economic losses for the farm.

Ultimately, they asked for full financial compensation for labor and administrative expenses, yield and profit losses, and material and leasing costs.

## 3. Discussion

Our research aimed to explore farmers´ hesitations and aspirations regarding their commitment to biodiversity within nature-protected areas and to elicit their motivation for implementing these measures. In our studies that used CATIs and online focus groups, the views of individual farmers were collected and analyzed together as the studies complemented each other.

The CATI evaluation of respondents´ answers from an economical point of view is challenging: many factors that have to be taken into account are not known and are applied differently by each farmer, not to mention fluctuating production prices and costs (this refers to direct as well as indirect drivers). The items and their amounts, e.g., in the calculation of contribution margins or in full-cost or partial-cost calculations, are not known. However, the task of this study was not to determine the extent to which farmers have "priced in" fixed costs in their answers. Moreover, the willingness of farmers to make their calculations transparent needs to be questioned. The fact that 25% of the farmers avoided this question feeds into this aspect. Furthermore, the farmers advocated for minimizing administrative expenses, as their perceived burden is immense. With regard to conceivable models for creating a financial incentive for biodiversity-friendly management, the farmers who participated in the CATIs unanimously advocated for a reward system. On the other hand, unlike the telephone-interviewed farmers, the farmers who participated in the online focus groups viewed the additional payment of success fees or bonuses critically. This is because they doubt the practical design of targets and performance controls, and they believe it would be challenging to implement these payments in view of the work in and with nature. The farmers stated that effectiveness would be difficult to measure, as other environmental influences are also relevant and a singular consideration of small areas is not possible, and not comparable.

In addition to purely financial incentives, the farmers named a number of other aspects that could motivate them to act in a more biodiversity-friendly way and which are almost entirely related to questions around the design of the agri-environmental measures and their cooperation with the administrative authorities. Three central, interwoven strands of action were identified.

1.  The need for the stronger appreciation of farmers´ contribution of agriculture to biodiversity, especially through monetary incentives.
2.  Flexibility and freedom in the implementation of measures in order to obtain the possibility of biodiversity-friendly land management.
3.  The recognition of farmers as partners in nature conservation through dialogue at eye level.

The in-depth online focus group study also revealed that the success fee favored by CATI participants was viewed questionably and skeptically in terms of implementation, as these participants could not envision a practical approach to evaluating success based on their experience of implementing existing measures. The current financial compensation for participation in biodiversity-enhancing measures based on action was mostly described by farmers as inadequate and was criticized for not reflecting the high value placed on biodiversity. In any case, there is currently no consideration of location and yield quality, or of additional expenses and increases of costs.

Our findings show that the issues faced by farmers practicing on arable land within nature-protected areas are many and varied. This is reflected in the heterogeneity and associated divergent viewpoints of farmers and is evident from both the CATI- and on-line focus group studies. The respondents´ answers regarding monetary issues, such as perceived monetary loss due to constraints in agricultural management, i.e., land use and perceived necessary monetary incentives, show that there is not only a large range within respondents' answers, but there is also a wide gap in the expectations around monetary incentives.

The aggregation of these studies show that our respondents are basically open-minded towards biodiversity-friendly measures and are willing to implement them, but expect sufficient financial support for these actions. They often stated that they would do more if they were financially rewarded. They also pointed out that promoting biodiversity must not be their sole responsibility. Various studies on farmers' willingness to act in a biodiversity-friendly manner have shown that farmers have a receptivity to change, but that acceptance of their work performance significantly affects them [45,46]. Furthermore, monetary incentives are not the only factor that plays a role. Other indirect influencing factors, such as policy and social frameworks, need to be considered.

The combination of qualitative and quantitative methods to investigate farmers' motivation towards enhancing biodiversity on arable land within nature-protected areas was suitable for our study. It enabled us to understand attitudes as well as motives, and we learned about farmers´ individual thoughts and behaviors. In this way, our analysis helps to gain a deeper insight into this area of study. However, the study is not representative as it only focused on farmers of existing nature-protection areas in Germany that were chosen through a project partner selection.

## 4. Conclusions

This study reveals that the individual farmer is deprived of determining their scope of action with regard to biodiversity, as the CAP mainly determines payments for agri-environmental measures in nature-protected areas, i.e., in the EU and the Federal States of Germany. This indirect driver (CAP regulations) has a high impact on farmers´ motivation to mitigate biodiversity loss. Monetary incentives are particularly important when we talk about agricultural land use within nature-protected areas. Especially in this study, incentives were found to be significant for achieving a high acceptance of well-considered measures, as farmers are willing to actively participate in agri-environmental measures aimed at effecting biodiversity, but their willingness would be increased by raising pay-

ments for desired public services [47]. To this end, it was demonstrated, that it is important to define comprehensible criteria that represents viable options for farms taking into account individual farm habitats. Offers made by regional contractual nature protection schemes are one approach to solving this problem [48].

Further scientific evaluations using the study methods should be carried out. Existing interdependencies and mutual dependencies of economy, ecology and society need to be further investigated to achieve the vital conservation of biodiversity.

**Author Contributions:** Conceptualization, A.T.; methodology, A.T.; validation, A.T. and W.T.; formal analysis, A.T.; investigation, A.T.; resources, W.T.; data curation, A.T.; writing—original draft preparation, A.T.; writing—review and editing, A.T. and W.T.; visualization, A.T.; supervision, W.T.; project administration, W.T.; funding acquisition, W.T. All authors have read and agreed to the published version of the manuscript.

**Funding:** The research for this article was funded by the German Federal Ministry of Education and Research (BMBF) as part of the project "Diversity of Insects in Nature-protected Areas (DINA)" and was handled by the VDI Project Management Agency (Grant Number FKZ 01LC1901).

**Institutional Review Board Statement:** Not applicable.

**Informed Consent Statement:** Not applicable.

**Data Availability Statement:** The generated dataset can be made available for research purposes upon request to the corresponding author.

**Acknowledgments:** We would like to thank Nicolas Fuchshofen for his valuable and appreciated contribution to the development of the sub-project of our work package "Institutional Framework and Stakeholder-Analysis", as part of the overall "Diversity of Insects in Nature-Protected Areas (DINA)—Project".

**Conflicts of Interest:** The authors declare no conflict of interest.

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
