# Peer review of "Hesitations and Aspirations of Farmers in Nature-Protected Areas"

_sustainability, doi:10.3390/su15043196_

Round 1

Reviewer 1 Report

This paper requires further development before publication. 

1. What is the research gap?

2. What are the research questions?

3. What are research hypothesis?

4. The description of CATI method should be added. Why did the authors choose the method of research.

5. The rules of division into study groups should be explained.

6. The paper does not include research results section. Please divide the second section into two parts. In my opinion from part 2.2.1 should be reserach results.

7. The research results shuld be compared with other authors results. Who did similar research?

8. The part discussion and conclusion should be divided into two parts.

9. The number of literature should be enlerged.

10. This paper has potential to be publish in the future after extensive correction and enlargement.

Author Response

Please see attachment and kind regards!

Reviewer 2 Report

Thank you very much for the review invitation of manuscript.

I read your paper “Farmers in German Nature-Protected Areas: Hesitations and Aspirations for the Commitment to halt the Loss of Biodiversity – an Assessment within the Project “Diversity of Insects in Nature-Protected Areas (DINA)” with great interest.

The manuscript's content seems to to be for a project report rather than a scientific article. The manuscript requires additional discussion with similar cases in the same or another country. More quantitative results than qualitative leaves are required. The manuscript is now almost a very good presentation of a farmer's problem in the study area.

The purpose of the manuscript is also not stated clearly in the manuscript. Therefore, the manuscript does not seem too to be publishable.

Furthermore, some mistakes in the manuscript must be corrected before the manuscript can be improved.

Title: check quotation marks

Line 12: There should be no citations in the abstract section

Line 12: “[…] what does authors mean?

Line 26: keyword famers: Farmer is a very common word, so consider whether or not to include it in your keywords.

Figure 2 conveys very little information, so it can be converted to text in 1-2 lines. All the Figures required higher resolution, and the text is difficult to read.

Need to explain why n = 97

Line 156: check mistake

When discussing expenses, the year must be mentioned

Line 234-239: this should be in the introduction section

To avoid being considered local, the discussion section should be expanded

Author Response

Please see attachment and kind regards!

Reviewer 3 Report

Review of Article:  

Farmers in German Nature-Protected Areas: Hesitations and Aspirations for the Commitment to halt the Loss of Biodiversity – an Assessment within the Project “Diversity of Insects in Nature-Protected Areas (DINA)

This is an interesting manuscript that deserves publication that is based on a mixed approach, following certain German experience of individual farmers by the means of a qualitative approach followed by a quantitative study. Meanwhile, the article can be improved following the bellow suggestions:

 ·         Title seems to be very long, it could like “Challenges and Obstacles of German Farmers within Nature Protected Areas towards Biodiversity Conservation” or rephrased in shorten form.

 ·         Ones you explain the role and sate of insects (lines 33-42), it is unclear the connection with topic, would be good to highlight the project objective and connection with published data with an additional sentence.

 ·         In the lines 41-42, you state:The endangerment of animal and plant species is 41 reflected by their classification in Red List categories”. This needs some references based on data generated in Germany or EU as example number of Vulnerable/Endangered/Critically Endangered assessed following IUCN criteria within last certain period. Further on, rephrase the sentence within line 39-41 and refer properly to the acronym IUCN.

 ·         Referring to the section “Methodology and its Findings” (line 98), it has to be clear thatMethods” are just behavior or tools used to select a research technique (in this case in your study), while the “Methodology” is analysis of all the methods and procedures of the investigation. So, it is recommended to use “methods”.

 ·         The sentence within lines 79-80 “The farmers’ inevitability to think economically is disturbed by political framework conditions set by EU (“Green Deal” [12] and “Farm to Fork Strategy” [13] ) and Germany”, need to be rephrased.

 ·         Under the section “Qualitative study - online focus groups 2 (third stage) - lines 214-227, it seems like discussion component, so this might be connected with following section of “Discussion”.

Author Response

Please see attachment and kind regards!

Round 2

Reviewer 1 Report

Authors corrected the paper. You can add more literature and publish the paper.

Author Response

Dear reviewer 1,

we appreciate your short feedback. We added some literature sources.
We look forward to processing publication.
Thank you!

Reviewer 2 Report

The manuscript has been quite well revised; however, Figures 2 and 3 could be improved in terms of clarity and font size.

Author Response

Dear reviewer 2,

we appreciate your short feedback. The improvements in clarity and font size of fig. 2 and 3 will be implemented in close coordination with the editor. 
We look forward to processing publication.
Thank you!

Reviewer 3 Report

I agree for procesing publication

Author Response

Dear reviewer 1,

we appreciate your short feedback.
We look forward to processing publication.
Thank you!